# COVID-19 and Public Health: Analysis of Opinions in Social Media

**DOI:** 10.3390/ijerph20020971

**Published:** 2023-01-05

**Authors:** Aleksey N. Raskhodchikov, Maria Pilgun

**Affiliations:** 1Moscow Center of Urban Studies “City”, 115280 Moscow, Russia; 2Interdisciplinary Research Laboratory, Russian State Social University, 129226 Moscow, Russia

**Keywords:** COVID-19, social media, public health, infodemic management, neural network technologies, perception of the situation

## Abstract

The article presents the results of research of public opinion during the third wave of the COVID-19 pandemic in Russia. The study touches on the attitude of citizens to public health, as well as the reaction of social media users to government measures in a crisis situation during a pandemic. Special attention is paid to the phenomenon of infodemic and methods of detecting cases of the spread of false and unverified information about diseases. The article demonstrates the application of an interdisciplinary approach using network analysis of texts and sociological research. A model for detecting social stress in the textual communication of social network users using a specially trained neural network and linguistic analysis methods is presented. The validity and validity of the results of the analysis of social network data were verified using a sociological survey. This approach allows us to identify points of tension in matters of public health promotion, during crisis situations to improve interaction between the government and society, and to timely adjust government plans and actions to ensure resilience in emergency situations for public health purposes.

## 1. Introduction

The COVID-19 pandemic has clearly demonstrated the importance of research and practical actions in the field of public health. Even the best, high-tech medical systems have proven to be ineffective in a pandemic situation. The spreading of COVID-19 to become a global phenomenon had a strong impact on the public health of various countries and increased the significance of the social media data that has been drawn upon in medical emergencies and crisis situations [1,2,3,4,5]. Doctors all over the world have become more likely to talk about the need for the formation of collective immunity and population health. It turned out that medicines and expensive equipment do not save people from the virus, good health and good immunity are much more important [2]. Thus, the long-term efforts of public health specialists have become more noticeable. Projects to combat bad habits, promote a healthy lifestyle, and efforts in the field of healthy urban planning have received more attention and support from citizens [2,5].

However, the COVID-19 pandemic also revealed a new negative phenomenon called “infodemic”, when hundreds of pseudo-experts and some public activists actively spread false information about the disease and incorrect treatment methods. The spread of false information created difficulties for doctors and authorities, some decisions on the introduction of sanitary restrictions were sabotaged because people did not believe in the seriousness of the threat. To combat this phenomenon, a special organization, the WHO Information Network for Epidemics, was created to study scientific methods and practices for preventing and detecting cases of dissemination of inaccurate and false information about diseases and methods of their treatment.

The sustainability of urban systems in a crisis depends a lot on the actions of the government. Responding to various public emergencies, the authorities take certain actions to manage dangerous and disruptive phenomena. Generally, there is a limited number of basic actions and tools that could affect the situation in a positive way [6]. It is worth noting that the actions of the authorities during the pandemic made great changes and created inconveniences in the daily lives of citizens. Such intervention, even if justified by the threat of an epidemic, inevitably causes an often-negative public response. The combined influence of two factors—the phenomenon of infodemia and the reaction of society to the actions of governments during the pandemic—has not yet been sufficiently covered by scientific research. The impact of the credibility factor, in particular, of interpersonal trust dynamics during the COVID-19 pandemic is shown in the research [7] with extensive use of big data and blockchain. The present study focuses on the impact that mass media and opinion leaders have on the users of social media. It also represents an attempt to study how people’s reactions depend on the degree of impact of emergencies on people’s daily lives and habits.

While analyzing social processes, it is essential to move away from the managerial approach that involves searching for an ideal solution that would suit the situation in the best possible way. In emergency situations that concern public health, the ability of governments to respond promptly to changes in public opinion becomes much more important. It is equally important to keep feedback channels open, the ability to respond correctly to criticism and disagreement of people and communicate with them in order to reach a compromise. Social networks provide an opportunity to quickly get information about people’s reactions to events. However, millions of messages on social networks require correct interpretation and reliable scientific methods of analysis. Social networks are full of false entities: chatbots, advertising and propaganda messages, and statements of pseudo-experts. All these factors are capable of generating thousands of automatic messages which significantly distorts network communications.

The analysis of users’ opinions about government services based on social media data and of users’ expectations towards the governmental structures has already been covered in scientific literature [1,3,8]. A software tool was developed comprising a collection of unsupervised Latent Dirichlet Allocation (LDA) machine learning and other methods for the analysis of Twitter data with the aim of detecting government pandemic measures and public concerns during the COVID-19 pandemic [9]. In addition, researchers have studied the role of social media and digital humor in challenging power and providing solidarity, as well as the role played by digital media during the pandemic, in the socio-political life of the Global South, as indispensable and revolutionary to human communication [10].

Representatives of many scientific fields have written about the effects of social media. For example, social media data is actively used in the area of Information Retrieval to solve various problems, for example, to determine the attendance of major events by users [11], to predict the general mood and attitude to a particular topic [12], and to develop recommendation algorithms that can be applied to decentralized social networks for community building [13]. Social media data is becoming an important source in the analysis of social processes, models of automatic extraction of behavioral characteristics, and perception of certain events and phenomena by citizens, since it allows real-time research on large data sets [14].

Network communications are an important component of modern media space. The specific features of inter-personal and intergroup communication are shown in numerous scientific studies. In particular, researchers note that social networks can filter actors, close groups, reduce the chance of unintentional communication, and isolate themselves from other groups, which causes some dosage and reduction of intergroup interaction of communities with different goals, political goals, interests, etc. [14,15,16,17]. User clustering in political discussions has been investigated in [18].

The creation of filter bubbles, echo chambers in groups with specific interests, leads to the isolation of intragroup communication, the strengthening of group relationships and norms, while at the same time it hinders communication between members of different virtual communities [19,20].

Numerous and diverse studies are devoted to the analysis of social and political polarization that occurs in various communicative processes in the digital environment [21]. On the other hand, the creation of filter bubbles, echo chambers in groups with specific interests leads to the isolation of intragroup communication, the strengthening of group relationships and norms, while at the same time it hinders communication between members of different virtual communities [19].

Christoph Stadtfelda, Károly Takácsbc, and AndrásVörös argue that traditional social network theories that are concerned with the evolution of positive relations (forces of attraction) are not sufficient to explain the emergence of groups because they lack mechanisms explaining the emergence of group boundaries. Authors find that a model that simultaneously considers forces of attraction and repulsion is better at explaining groups in social networks [22]. Meanwhile, some social and systemic features of social networks contribute to intergroup communication even more than offline or traditional communication channels. Scott A. Golder and Sarita Yardi show that two structural characteristics, transitivity and mutuality, are significant predictors of the desire to form new ties [23].

The specificity of the information dissemination in social networks has already received a multi-dimensional coverage in scientific research [24,25].

Obviously, the most attention in social media analysis during the COVID-19 pandemic is focused on the strengthening of the healthcare system [26,27,28,29].

The COVID-19 IMPACT project studies behavioral patterns that constitute either risk factors for being infected by (or spreading) the virus or consequences of risk behaviors: “The International COVID-19 IMPACT study is based on theories of psychology and aims to provide evidence on the mechanisms people use to cope with the impact of COVID-19, such as physical isolation, and on people’s health behaviors” (https://ucy.ac.cy/acthealthy/en/covid-19-impact-survey (accessed on 27 November 2022)). An international online opinion poll was conducted in 78 countries and regions as part of that project.

Several text corpora aimed at the analysis of data related to the spread of the coronavirus infection have been formed, for instance, The Coronavirus Corpus by Mark Davies and Sketch Engine.

The Coronavirus Corpus by Mark Davies contains downloadable, full-text corpus data from ten large corpora of English: iWeb, COCA, COHA, NOW, Coronavirus, GloWbE, TV Corpus, Movies Corpus, SOAP Corpus, Wikipedia, as well as the Corpus del Español and the Corpus do Português. Corpus is designed to be the definitive record of the social, cultural, and economic impact of the coronavirus (COVID-19). The corpus (which was first released in May 2020) is currently about 1344 million words in size, and it continues to grow by 3–4 million words each day (https://www.english-corpora.org/corona/, accessed on 1 January 2022).

The COVID-19 corpus consists of texts that were released as part of the COVID-19 Open Research Dataset (CORD-19). Reference: COVID-19 Open Research Dataset (CORD-19). 2020. Version 2020-05-02. doi:10.5281/zenodo.3715505 (https://www.sketchengine.eu/covid19/, accessed on 1 January 2022).

Those resources provide ample opportunities for scientists to research various aspects of the spread of coronavirus as well as the perception of the pandemic by diverse groups of actors and the consequences of it for them. 

At the same time, the Russian language is usually outside the context of such studies and is not included in international scientific resources. In this study, we present the case of using both neural network analysis of Russian-language data and standard sociological research procedures, which strengthens its validity. Neural network technologies allowed us to analyze both implicit and explicit information as well as assess actors’ hidden reactions and opinions that shape their perceptions of the research issues.

The importance of social media in the recovery of sustainable development of various sectors during the pandemic has also been covered by multiple research articles. Thus, it has been proven that large technological platforms are ones of the few beneficiaries of the global economic crisis caused by COVID-19, mainly because of the evolution of consumption patterns involving digital tools (basically, social media) [30]. Social media play a huge part in the lives of children, adolescents and students, especially during the COVID-19 pandemic, when a PC is not only a means for entertainment and leisure purposes, but also a necessary tool for studying and communication with other people in their daily lives. The article [31] presents complex recommendations for using social media as means for young people’s wellbeing during the pandemic. The article by Steven Ratuva, Tara Ross, Yvonne Crichton-Hill, Arindam Basu, Patrick Vakaoti, and Rosemarie Martin-Neuninger presents a comparative analysis of how communities have developed people-based resilience in response to the global impact of COVID-19. The authors explore some of the ways in which local communities have mobilized their cultural resources to strengthen their social solidarity and mitigating mechanisms against the continuing global calamity [32]. The analysis of tourists’ communications behavior on the Internet allows city managers to obtain correct information on people’s needs and expectations, to respond adequately to them and to ensure effective governance and promotion of their city on the global scale [33].

The examples above show that social media data can be used to study the transformations caused by the pandemic in multiple areas of the society. However, little attention has been paid to people’s and communities’ reactions to the actions of the authorities during the pandemic. Yet it is an issue of major importance, since it helps to understand which actions have caused the most dissent and resistance among the citizens. It is necessary to take this into consideration while making decisions in emergency situations, which unfortunately have been quite frequent lately.

In crisis conditions, social sustainability is also of particular importance, which focuses on maintaining and enriching the well-being and quality of life of contemporary and succeeding generations [34,35,36,37,38,39,40,41,42].

The third wave of the pandemic in Russia was the most severe in large cities, especially in Moscow. The article presents the results of a study of citizens’ reactions to the actions of the Moscow authorities and of changes of the relevant information agenda in the digital environment during the third wave of the COVID-19 pandemic in Russia.

An interdisciplinary approach was followed in this study, which consisted of two stages. At the first stage of the study, we analyzed digital data, and at the second stage of the study, an offline sociological survey was conducted.

The purpose of the study was the analysis of the information media environment and communications behavior of actors during the third wave of the COVID-19 pandemic in Russia. The study also aimed to assess citizens’ perception of government actions.

The hypotheses involved an interdisciplinary approach, which consist of two stages: analyses social media data and an offline sociological survey allows the most correct analysis of the development of the social situation the method involving social stress assessment in user generated content allows to assess users’ reactions by the number of messages and their sentiment as well as understand the tensions that any given issue can create in the society. The analysis of citizens’ reactions on social media helps determine which actions of the authorities turn out to be the most distressing for people and can provoke mass dissent.

## 2. Materials and Methods

### 2.1. Materials 

Data from corpora, social networks, video hosting services, instant messengers, microblogging, blogs, reviews, forums, thematic portals, online media, print media, and TV news stories related to the third wave of the COVID-19 pandemic in Russia have been used as the materials for the study.

According to WHO, the third wave of the COVID-19 pandemic in Russia lasted from early June to late September 2021 (Figure 1).

We chose analysis for the period between early June 2021, when the number of new cases of COVID-19 grew rapidly and the first anti-COVID-19 measures were introduced, and mid July 2021, when the number of infections began to decrease, and the Moscow authorities lifted the restrictions.

Data collection period: 2 June 2021 to 16 July 2021.Number of messages: 176,334Number of actors: 103,018.Engagement: 712,560.Audience: 581,905,792.

Since the subject of our study is the reactions of Internet users to the actions of the authorities, of all messages published during the relevant period, we picked the ones related to governmental actions. Chronologically, the most important actions taken by the Moscow government included declaration of “non-working days”, introduction of quarantine restrictions (closure of food courts and children’s playrooms in shopping malls, nighttime curfew for cafes and entertainment enterprises, restrictions in parks), distance learning for schools and universities, mandatory vaccination for employees in certain economic sectors, and promotion of voluntary vaccination by means of a valuable prize drawing and introduction of QR codes for visiting cafes and restaurants.

Therefore, for the purpose of our study, six text corpora were formed containing messages and comments published on social media sites. We also included statements by public officials, mass media publications, as well as posts by popular bloggers and various Internet users.

All messages of the dataset were clustered in 132 “topic groups” based on the textual matches (Appendix A). From those topics, the ones related to one or several of the above anti-COVID-19 measures were selected by means of the textual content analysis. Thus, six datasets were singled out comprising users’ comments to the following measures: non-working days, quarantine restrictions, vaccination prize drawing, distance learning for universities, mandatory vaccination, and QR codes (see Table 1). 

Choosing a digital platform for discussions, actors tended to prefer social media sites (904,045,226), video hostings (177,729,334), and microblogs (99,856,257). VKontakte tops the list of the most-preferred social media (692,861,581), followed by YouTube (177,729,334) and Facebook (120,285,422) (Figure 2).

The content was mainly comprised of comments (585,818,027) and posts (568,581,707) (Figure 3)

### 2.2. Methods

An interdisciplinary approach was followed in this study. 

At the first stage of the study, we analyzed digital data. We used neural network text analysis, sentiment analysis, analysis of lexical associations, and content analysis.

The data were collected using Brand Analytics algorithms (https://br-analytics.ru/, accessed on 1 January 2020).

To achieve the objectives of the study, a neural network text analysis was conducted. Neural network technology TextAnalyst 2.3 (https://www.analyst.ru/, accessed on 1 January 2020) allowed us to identify and analyze the topic structure of the database, identify the semantic network and the associative network, analyze the core of the semantic network and word associations, as well as make a summary [43]. 

We also used the SketchEngine software [44,45]. 

The content analysis [46,47,48] was performed by means of the AutoMap service (http://casos.cs.cmu.edu/, accessed on 1 January 2020). 

The sentiment analysis was performed by means of the Eureka Engine sentiment determination module (http://eurekaengine.ru/, accessed on 1 January 2020).

For visual analytics, the Tableau platform was used (www.tableau.com, accessed on 1 January 2020).

Analysis of users’ emotional reactions to the actions of the authorities by means of information stress assessment in text messages is a major part of this study.

For that purpose, sentiment analysis and associative network analysis were used. Associative search, associative network analysis and the Word Association (WA) method allowed us to expose implicit information in the content generated by users and identify reactions that are the most precise indicators of citizens’ attitudes towards the actions of the authorities during the third wave of the COVID-19 pandemic.

The WA paradigm analysis has a long and multifaceted history [49,50,51]. By means of implicit association tests (IAT), it helped study implicit social cognition, subconscious motivations, attitudes towards the presented stimuli, as well as automatic associations of subjects who prefer not to demonstrate them at a conscious level (see, for instance, Project Implicit (https://www.projectimplicit.net/, accessed on 1 January 2020). The potential of associations in the analysis of various types of network data has also already shown its effectiveness [52,53]. In this study, the analysis of lexical associations was performed on the material of the clusters identified in the course of the sentiment analysis using neural network technologies [54], which made it possible to draw conclusions about the preferential perception by the users of the actions of the authorities, to identify the most frequent associations that characterize the actors’ attitude towards the subject of analysis.

A detailed description of the author’s methods for analyzing digital content is presented [43,55].

At the second stage of the study, an offline sociological survey was conducted. The survey “Moscow citizens’ attitudes towards public health and the coronavirus situation” was conducted from 20 September 2021 to 17 October 2021 by Stolitsa Social Research Agency (ASIS) at the request of the Moscow Center of Urban Studies “City”. The data were collected by means of a telephone survey (a formalized interview). The survey involved 1200 respondents who reside permanently in Moscow and are over 18 years old. The study used stratified sampling based on age and gender criteria, which were formed on the basis of statistical data provided by the Federal State Statistics Agency ROSSTAT. With a confidence interval of 95%, the sampling data error could be no more than 5%. The survey was conducted after the end of the third wave so as to avoid data corruption due to potential emotional reactions of the respondents caused by the crisis situation.

### 2.3. Procedures

#### 2.3.1. Digital Content Analysis

Design of digital content analysis is presented in Figure 4.

#### 2.3.2. Offline Sociological Survey

Design of offline sociological survey is presented in Figure 5.

## 3. Results

### 3.1. Key Topics 

The topic modeling of the selected messages and main data set topics shows that vaccination was the issue most widely discussed on the Internet. During the first wave of the pandemic, the proportion of messages related to vaccination barely reached 3% of the total array of messages dedicated to the COVID-19 pandemic [55]. During the third wave of the pandemic, 75 out of 132 topics were already related to the discussion of vaccines, and almost 60% of all messages related to vaccination.

These were the topics that attracted the most attention:

1.Mandatory vaccination for students (8879):Students have to get vaccinated before the start of the academic year, preferably before July 1, Minister of Health said.Yeah, and they said it was voluntary… Schoolkids, you’re next! Users’ reactions: “You’ve got to be really dumb to think it’s going to be voluntary :)VOLUNTARY, oh yeah babyWell, it is, but if you’re not vaccinated you can’t go anywhere or do anything”2.Routine medical care for vaccinated only (4276):Routine medical care in Moscow will be provided only to vaccinated citizens (ria.ru) Users’ reactions:“*gasp* At least they’re not killing us… yetWell, you can relax for now, it says there that emergency aid’s going to be provided to unvaccinated, too”3.Discrimination against unvaccinated people (3875):Peskov made a comment about the discrimination of unvaccinated people:“The reality is that such discrimination is unavoidable…”And in a few hours all hell broke loose.Sobyanin made a statement on the new restrictions in Moscow:As of June 28, only the persons who have either been vaccinated, or have recovered from COVID-19 within the last six months, or have a negative PCR test valid for three days will be able to visit public catering establishments.Fan zones and dance floors will have to close, too, if the venues fail to arrange “COVID-19 free zones” available only for those with QR codes.As ombudsman of the restaurant business Mironov said, Moscow cafes and restaurants could lose up to 90% of their visitors because of those new restrictions (…).Also, unrest grows over mandatory vaccination that has recently become fact in 12 regions of Russia.Users’ reactions: Wow, and what about our freedoms and rights?! How typical! And as ever, Lenin and Stalin are to blame, right?And may I ask Mr. Peskov, do we still have the Constitution or not? How is discrimination against unvaccinated in line with the constitutional rights of citizens?

The second feature of the media environment during the third wave was that unofficial information absolutely prevailed. The number of messages distributed by various Internet communities and bloggers greatly outnumbered those published by the authorities’ official channels and mass media. The share of such unofficial messages in the data set is over 73% of the total; 90 of 129 topics were initiated and distributed through by unofficial channels.

Notably, it was the news that tourist travel to Turkey had been resumed that reaped the greatest number of messages (22,656):

"Turkey is open again, yay! And the non-working days are off.Mass events no, football yes.Zoos in the open no, cinemas indoors yes.And our authorities and scientists are still not sure if the vaccination is voluntary or mandatory, if it is effective at all and against which variant, and how many antibodies do we have to have.”as well as the information that coronavirus was man-made (11,057):“The truth the head epidemiologist has been hiding about the artificial origin of coronavirus.Chinese scientists compiled the available data and concluded that SARS-CoV could be man-made.It’s a global conspiracy against common people. I wish you all health; take care of your friends and loved ones).”

### 3.2. Internet Users’ Reactions to the Actions of the Authorities

One of the objectives of the study was to assess the reaction of users to the actions of the authorities during the third wave of the pandemic. Quantitative analysis of messages in social networks made it possible to identify six main ones that received the most discussion: (1) mandatory vaccination, (2) distance learning for universities, (3) announcement of non-working days, (4) quarantine restrictions, (5) introduction of QR codes, and (6) drawing of prizes among participants of vaccination. The criterion for choosing these topics was the largest number of messages and comments from network users. These topics became the basis for the formation of 6 corpora of texts and their subsequent semantic analysis. To study Internet users’ reactions to the actions of the authorities, we used sentiment analysis, identifying three clusters: positive, negative, and neutral. The analysis shows that neutral messages make up a major part of the total (996,903,716), negative ones are an insignificant number (125,112,033), and positive ones amount to even less (68,112,910). VKontakte tops the list of source sites with the greatest number of negative messages (51,061,487) (Figure 6).

We also conducted an associative search, identified the associative network, and analyzed the WA (Figure 7, Figure 8, Figure 9, Figure 10, Figure 11 and Figure 12). 

Mandatory vaccination—(10/22 758)Reaction contents:(…) this is beyond good and evil obviously this is a breach of the Constitution and federal acts such measures such norms compulsion to vaccinate really wrongful contrary to the Article 41 of the Russian Constitution and to the second federal act on preventive immunization any vaccination can be only voluntary only upon the informed consent of the citizen nobody has the right no state no government no ministry of education no prime minister nobody has the right to coerce us to injections coronavirus vaccination is not on the national immunization schedule (…). 

Distance learning for universities—(10/4 473)Reaction contents:(…) Applicants not vaccinated against COVID-19 can be barred from enrolling in universities. Alexander Bashkin, member of the Federation Council, your typical party hand, said that applicants not v*ccinated against COVID-19 and not having QR codes could be barred from enrolling in universities and colleges. As for the students, they could be banned from going to classes: for them, it would be either academic leave or distance pseudo-learning, or expulsion! Apart from everything else, this is a breach of international treaties that explicitly prohibit to discriminate against those who did not get vaccinated because of potential health risks or because they did not want to (…)

Non-working days—(10/4339)Reaction contents:(…) The legal sense of those "non-working days” paid by employers in the spring of 2020 was unclear, too, to say the least. But the reason for continuing large-scale administrative improvisations then and now is our government’s unwillingness not only to comply with their own laws, but also to deal with the consequences of their actions.It is not so hard to draw up a national plan. To write new federal laws and make the vaccination campaign legal is even less of a problem. But then they will have to promise things to people and report on how they’ve kept their promises. How many millions of Russians will be vaccinated in June? In July? In August? How quickly are they planning to vaccinate those so-called groups at risk, including elderly people? What prizes and stimulations are those who will get vaccinated going to get? What federal restrictions could be imposed on those who will not get vaccinated and what restrictions could be lifted for those who will? (…)

Quarantine restrictions—(10/4400)Reaction contents:“Warn your employees in advance that you will have to suspend those who were supposed to get vaccinated, but did not do so. Explain to them that there is a clear deadline. For instance, in Moscow, the first shot is to be made before July 15. As of June 16, you will not be able to admit them to work unless they get vaccinated or the authorities lift the restriction. If your employees say they refuse to get vaccinated, draw up an act and a suspension order.” Once again, dear author of this piece, although you are not dear to me at all, on what basis is the employer to draw up a suspension order? The Ministry of Health thinks that we will not be able to reach the so-called herd immunity before the mass vaccination of children starts, and until that time they intend to keep in place the state of emergency and all the anti-epidemic restrictions, including obligatory masks everywhere and at all times.

QR codes—(10/575)Reaction contents:(…) Those who are for the mandatory vaccination will bring about a disaster for the Russian economy and the wellbeing of our citizens. We will either have to get injected with untested vaccines, get segregated and disenfranchised, live with their unlawful QR codes and other examples of “new normality”, or businesses will go bankrupt (…).(…) It is really a Morton’s fork: either we will be segregated and disenfranchised and will have to live with QR codes and other examples of “new normality”, or businesses will go bankrupt… And there are also small enterprises, QR codes will kill them. It seems that all goes according to plan. Let me remind you what Ida Auken said at the World Economic Forum: “Welcome to 2030. I own nothing, have no privacy, and life has never been better.” Instead of going out on strike, instead of opposing the unlawful measures of Rospotrebnadzor and municipal authorities, as restaurant businesses do in Europe, our businesses obediently go to the slaughter.

Vaccination prize drawing—(10/38)Reaction contents:The authorities intend to hold a prize drawing of five cars worth about one million rubles each week. The results will be announced on Wednesdays, starting on June 23, on Moscow 24 TV channel, Mayor of Moscow said.And how much is the monthly subscription for going out, having some air and quality time on a bench in a park? :deep in thought:Will there be any discounts for pensioners and elderly people? You know, if they get tired while they walk to a park to have some, you know, fresh air (God forbid, but what if it is their last walk, with their chronic diseases progressing and the wear and tear this fun and eventful life puts upon them)And such are the people around us! Yeah, the ones that walk our streets right now. The authorities arranged a prize drawing of cars, for God’s sake. It’s like when you buy a vacuum cleaner, and they also give you a mini oven. Marketing, that is. Come on, guys, when you get vaccinated, your life and wellbeing are your prize and not a car.

Assessing the users’ reactions to the actions of the authorities, it is worth bearing in mind that some of the measures were already familiar to Muscovites since they had been in force during the first and the second waves. For instance, non-working days and distance learning for universities had already become a common practice and did not provoke many discussions on social media. However, restrictions in parks, which also had to have been already familiar to citizens, caused a negative response since they applied to all park facilities, including benches. Multiple users found this absurd, especially with regard to elderly people who at times need to have a rest while they are taking a walk. Notably, the authorities took the negative reaction into account and later lifted that restriction.

The most widely discussed and the most condemned by the users were the new measures, in particular, mandatory vaccination for workers and employees in certain economic sectors, suspension from work for those who did not wish to get vaccinated, and QR codes/PCR tests for visiting cafes, restaurants, theatres, cinemas etc. The mandatory vaccination decree looked relatively lenient on paper, since it did not apply to all organizations, but only to the ones active in particular economic sectors (retail, food service, public transportation, housing and utilities, consumer services, beauty salons, sports and entertainment, education, childcare, as well as financial institutions and public offices), and the vaccination rate among workers and employees had to be only 60%. In fact, that decree affected broad segments of the population, provoked heated negative reactions on the Internet (there were messages about employers coercing their employees to get vaccinated under the threat of dismissal) and triggered numerous petitions against mandatory vaccination and public pleas to call it off. Widespread purchases of forged vaccination certificates could be considered yet another type of public reaction to that measure.

Muscovites showed their dissent, in particular, by inventing a portmanteau word Moscauschwitz (Moscow + Auschwitz):

(…) And then Kanalgeruch (Sergey Sobyanin’s real name according to conspiracy theorists) will say that all… is more contagious than the Indian variant, and that the citizens of Moscauschwitz have to abandon the city for their own benefit, since… air, water and food… According to preliminary data, Berel Lazar, chief editor of… will coordinate the evacuation (…).

The discussions of new restrictions on visiting cafes, restaurants, and culture and entertainment venues were no less heated. That measure triggered multiple negative messages on social media as well as accusations of restrictions on people’s rights and freedoms and discrimination against unvaccinated citizens, fueled popular fears and doubts as to the safety of vaccines, and prompted rumors that people got microchips implanted under their skins instead of vaccines. Posts mentioning Saint John the Apostle’s apocalyptic prophesies about the “seal of Antichrist” were also widespread on the Internet. Judging by the fact that there were reports of queues in vaccination centers after that particular measure had been publicly announced, it can be assumed that restrictions on visiting cafes and restaurants turned out to be the most effective incentive to vaccination.

### 3.3. Results of the Sociological Survey

Nevertheless, analysis of posts, messages and digital traces on social media does not always help to get a clear picture of the public opinion. This has to do with the fact that the typical subject of social media analysis is users’ communications, including messages initiated by actors, and the ensuing discussions. It is methodically incorrect to equate communications on the Internet with the real public opinion due to numerous factors (actors’ own goals and motives, the fact that only a limited number of people takes part in discussions on the Internet and that the samples are not representative, etc.). Social media are more of a something like signal system that notifies us of emerging problems, trends and especially conflicts, which are quickly detectable on social media sites.

To assess the public opinion on the events of the third wave of the pandemic, the authors of this article conducted a sociological survey in Moscow to find out how the citizens perceived the pandemic and the vaccination process. 

The survey was aimed at discovering the respondents’ attitudes towards various actions taken by the authorities during the pandemic. They were offered a list of twenty items that included support measures for certain categories of citizens and businesses, quarantine measures and restrictions, and mandatory vaccination (Figure 9; the diagram shows the percentages of “supporters” and “non-supporters”; the “no answer” share is not shown for visual convenience, which is why the total of both responses is less than 100%). The results show that the level of acceptance of support measures for various social groups in Moscow is relatively high. The least popular measures were restrictions on visiting parks and squares (78.5%), closure of cafes, restaurants and food courts (71.2%) and closure of children’s play areas in parks and shopping malls (59.4%). Most of the respondents also disapproved travel restrictions (54.2%) and QR codes in cafes and restaurants (51.5%). The findings do not fully match the results of the semantic text analysis. The most negative reactions on social media during the third wave were caused by the closure of cafes, restaurants and food courts and later by the introduction of QR codes for visiting them. Those are the most “not supported” measures of the survey, too. By contrast, the mandatory vaccination for employees of certain economic sectors, heatedly discussed by Internet users, was not unanimously condemned by the respondents of the survey. On the contrary, most of them said that they supported that initiative of the authorities (57.2%). Therefore, vaccination could be an issue that causes great argument and splits the public into two opposing parties. That is why it is so broadly discussed on the Internet. There is another explanation, though, implicating deliberate politization of that issue in Internet discussions (Figure 13). 

Question: Which anti-COVID-19 measures taken by the Moscow government do you support and which ones do you consider unreasonable?

The data obtained by the survey shows that the citizens’ attitudes towards vaccination are ambiguous. Less than a half of the respondents (46.2%) are in favor of it and expect the pandemic to end soon due to mass vaccination. In contrast, nearly one third (36.4%) are skeptical towards the vaccination campaign, arguing that the vaccines have not been properly tested and their benefits have not been confirmed. The share of those who strongly oppose vaccination is relatively small (10.1%); they are concerned that the vaccines could harm people’s health.

During the survey, the respondents were asked an additional question: Where do you get the most of the information about coronavirus? The answers show that the respondents’ attitudes towards vaccination tend to depend greatly on the source they get their information from. Breaking down all information sources into official (TV, printed media and radio) and unofficial ones (blogs, social media and opinions of relatives and neighbors), it becomes evident that the respondents who prefer to get their information from the official channels are more likely to be in favor of vaccination. On the other hand, the ones who have more trust in social media and the opinions of their relatives and neighbors are more likely to be skeptical about vaccination and fear that it could cause harm to their health (Figure 14). 

Question: What is your attitude towards vaccination against COVID-19? 

Although most of the respondents (56.5%) recognize the threat the pandemic poses, there is still quite a large share of those who think that the risks are somewhat exaggerated (32.5%), as well as of those who believe that COVID-19 is no more dangerous than a seasonal flu (7.9%). Notably, there is a strong correlation between the respondent’s attitude towards COVID-19 and their subjectively perceived health condition. Those who consider themselves to be in good health tend to underestimate the risks of the infection, and vice versa (Figure 15). 

Question: What is your attitude towards the risks the COVID-19 pandemic presents? 

The pandemic and the economic crisis it caused affected labor force greatly. If we consider changes that has happened to those employed, the clearest trend appears to be transferred to remote or partly remote work. Thus, the survey shows that one in five employed respondents has started working from home. Women and young people of 18 to 34 years are more likely to be in that group. In addition, 17.4% of the employed respondents said that they had lost their job due to the pandemic or had had to change it. Notably, people aged 45 to 55 were more likely to lose their job, while working pensioners tended to change it. It is also noteworthy that the share of those who perceive the financial situation of their family as one close to poverty (i.e., they can afford to buy food, but purchasing new things or clothes could be a problem) is especially large among the respondents who have lost their jobs due to the crisis caused by the pandemic (Figure 16). 

Question: How has the COVID-19 pandemic affected your job (excluding unemployed respondents)?

In the course of the sociological study, the respondents’ opinion on the factors that have the greatest impact on human health was also clarified. The results obtained demonstrate that people are well aware of the importance for health of such factors as lifestyle, nutrition, the presence of stress, heredity, bad habits and sports. The data obtained largely correspond to the Dahlgren and Whitehead model of public health.

## 4. Discussion

The analysis of the actions taken under the governmental health policy in order to control the spread of COVID-19, as well as public reactions to the anti-COVID-19 measures of the authorities have already been presented in scientific research papers [1,3,5,56]. During a pandemic, people’s behaviors are crucial in the process of stemming the transmission of the disease. This study fully confirmed the results that were made in previous studies regarding the two main types of communicative behavior of residents during a pandemic. Already at the beginning of the COVID-19 pandemic, citizens have been showing two types of reaction patterns. One is characterized by increased anxiety and fear, which cause the individual to strictly observe all anti-epidemic measures and may sometimes lead to panic. The other is distinguished by firm rejection of all preventative measures and by protests caused by the individual’s denial of the risks that the disease may pose [57,58,59,60,61].

In research of this kind, it seems appropriate to pay special attention to the fact that for the most part, the information circulating on social media sites is reposted from mass media and that the reactions of the social media users are only a part of the common database (text corpus). That is why it is so important to distinguish between natural reactions of social media users and artificial ones (the so-called “information waves” caused by critical publications on mass media and statements of various politicians). The ratio analysis of such natural and artificial reactions is of great methodological interest in itself. The authors will present a case study in their upcoming research articles.

## 5. Conclusions

In this study, we analyzed the reactions of Moscow citizens to the actions of the authorities during the third wave of COVID-19 in Russia on social media and the dynamics of the related information agenda on the Internet. The results indicate that the vaccination promotion initiatives attracted the greatest attention. Coercive measures, such as mandatory vaccinations of certain categories of employees and restrictions on visiting cafes and restaurants for unvaccinated citizens, were met with the most dissent. Even in an emergency, measures based on prohibition and compulsion trigger strong negative reactions among citizens. Although such attitude could hardly be called rational, it is necessary to take into account when drawing up an action plan, that in people’s minds, emotions often come before reason.

In the decision-making process, small mistakes can sometimes bring about powerful adverse consequences. Thus, the ban on the use of park facilities drew sharp criticism on social media. Obviously, the authorities only intended to minimize mass gatherings in parks. However, due to a misunderstanding or an honest mistake, an excessive action was taken, prohibiting, among other things, the use of park benches. It was not taken into account that elderly people use benches when they need some rest during their walks and that that would not have prevented them from keeping physical distance from each other.

The analysis of posts on social media shows that the restriction on visiting cafes and restaurants turned out to be the most effective vaccination promotion measure. There were reports of queues in vaccination centers and even shortages of vaccines after that particular measure had been publicly announced. City dwellers, used to having unlimited access to the so-called third places, were not prepared to give up the possibility to go to a café or a restaurant, and although that measure was sharply criticized on social media, too, it worked quite well. However, mandatory vaccination of certain categories of employees and suspension for those who refuses to get vaccinated were an entirely different thing. That measure seems to have yielded no significant results. In fact, it triggered multiple protests, including petitions against mandatory vaccination, and accusations of civil rights violation.

Thus, all hypotheses were confirmed.

An interdisciplinary approach was used which consisted of two stages: analyses of social media data and an offline sociological survey allowed correct analysis of the development of the social situation.

The study showed that the analysis of citizens’ reactions on social media helps determine which actions of the authorities turn out to be the most distressing for people and can provoke mass dissent.

Data analysis confirmed that the method involving information stress assessment in user generated content allows us to assess users’ reactions by the number of messages and their sentiment (positive, negative, or neutral) as well as understand the tensions that any given issue can create in the society. That is a qualitative characteristic that helps, among other things, to indicate potential conflicts.

The comparison of data obtained by means of text analysis and the sociological survey shows that users’ reactions on the Internet do not always give a clear picture of the public opinion. Thus, numerous negative comments on social media related to non-working days, distance learning for schools, and universities and mandatory vaccination for certain groups of employees find little support in the general public—if the sociological survey is to be believed, since most respondents were in favor of such measures. At the same time, restrictions on the work of various organizations and businesses, which did not garner widespread attention on the Internet, were condemned by the respondents of the survey.

The results indicate that the authorities should not focus exclusively on the criticisms they get on social media. In some cases, it is also useful to conduct sociological surveys to get a clearer picture. Still, although social media data do not always reflect public opinion as it is, they could indicate new trends and incipient conflicts. A sound assessment of such trends by means of emotional stress analysis helps predict the development of potential conflicts and take preventative compromise measures. 

This approach, based on the assessment of social media users’ reactions to the actions of the authorities, is instrumental in preparing a timely response to growing social tension and emerging conflicts. The procedures used by the authors, including analysis of news stories on social media, emotional stress analysis of text messages, and diagnostic sociological surveys, helps assess the efficiency of measures and detect mistakes during the spread of diseases for public health purposes.

The study showed that in the information society, issues of interaction with public opinion leaders are of great importance. In crisis situations, they are able to settle social conflicts at the expense of their authority, and vice versa, to deliberately incite public discontent with the actions of the rulers.

The COVID-19 pandemic has shown the importance of public health formation to combat the spread of viruses within the population. However, the infodemia factor has demonstrated how many false opinions and pseudo-expert judgments about a healthy lifestyle, nutrition, and behavior are spread on social networks. The method of detecting social stress in text messages in social networks presented by the authors provides opportunities for qualitative analysis of social network data. Fixing the growth of stress in network communication can be used not only in crisis situations, but also in normal times to identify social conflicts and spread false information on public health issues.

## Figures and Tables

**Figure 1 ijerph-20-00971-f001:**
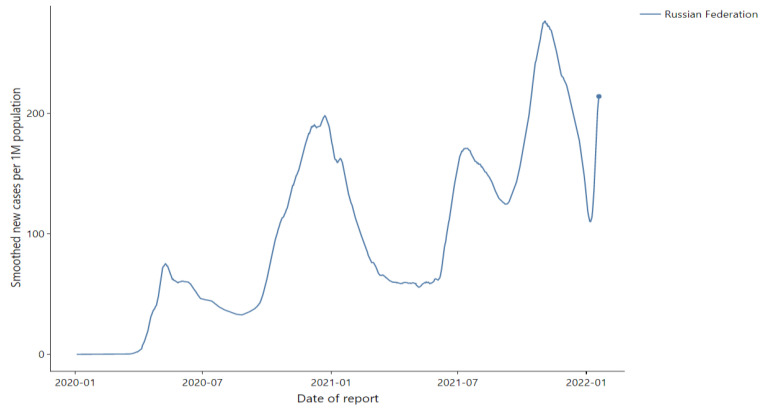
Spreading of the COVID-19 pandemic in Russia. (https://worldhealthorg.shinyapps.io/covid/, accessed on 30 January 2022).

**Figure 2 ijerph-20-00971-f002:**
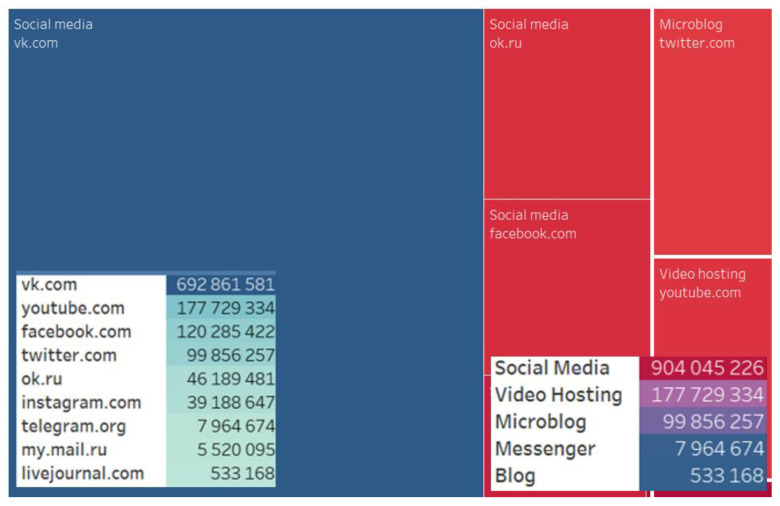
Digital platforms and social media types.

**Figure 3 ijerph-20-00971-f003:**
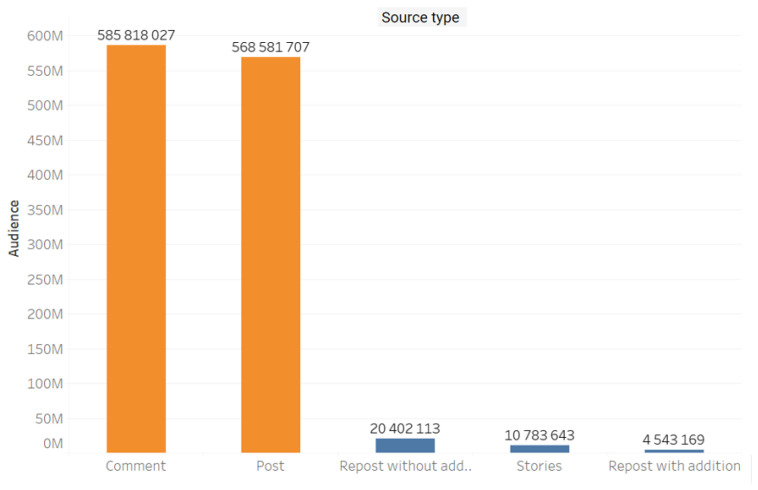
Message types (post, comment, repost with addition, repost without addition).

**Figure 4 ijerph-20-00971-f004:**
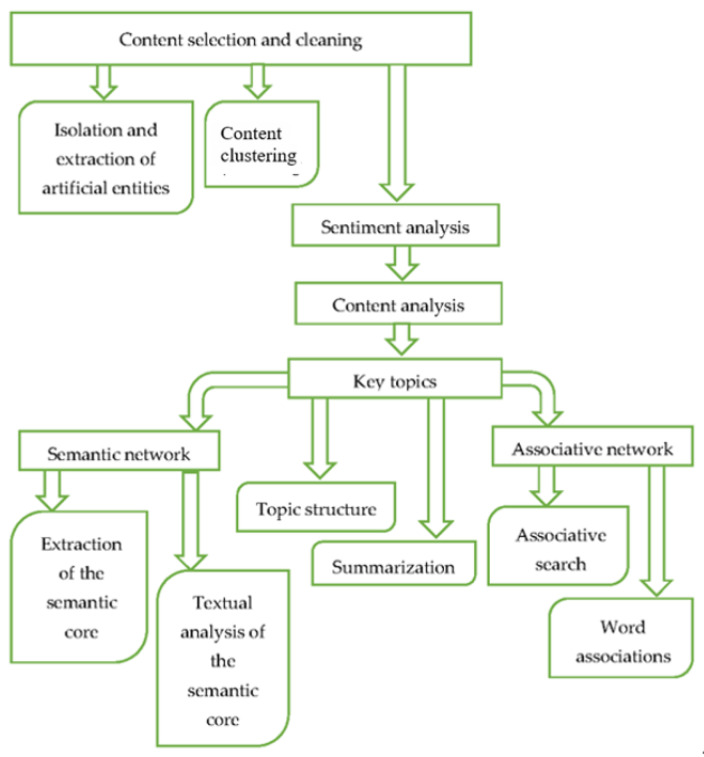
Design of digital content analysis.

**Figure 5 ijerph-20-00971-f005:**
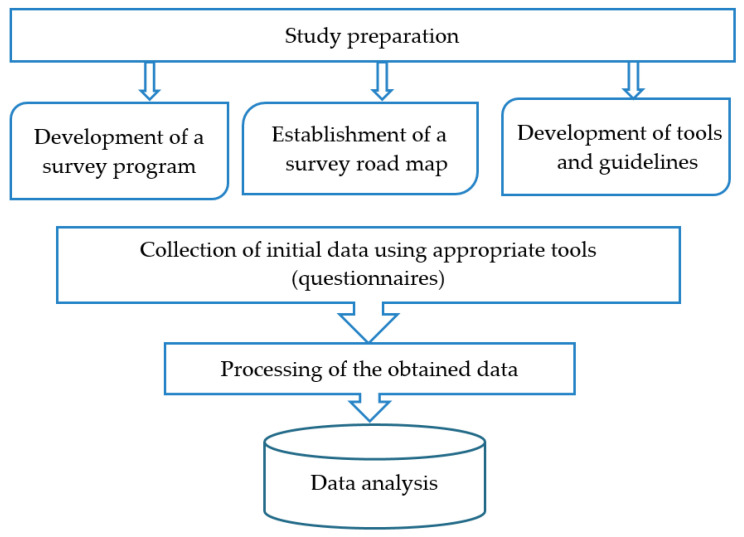
Design of offline sociological survey.

**Figure 6 ijerph-20-00971-f006:**
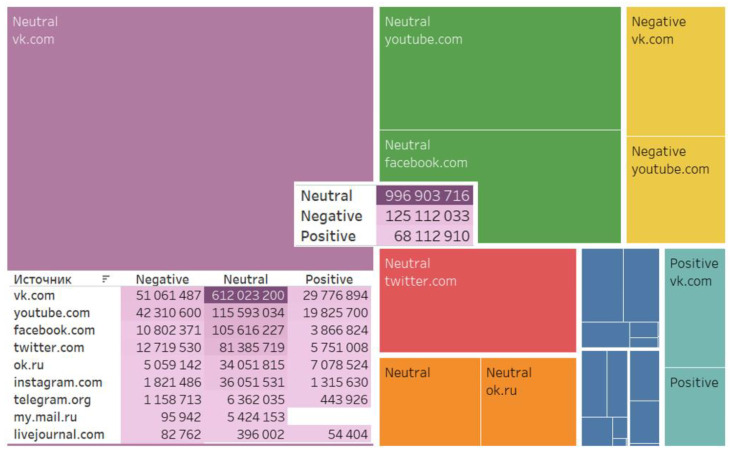
Sentiments of the content.

**Figure 7 ijerph-20-00971-f007:**
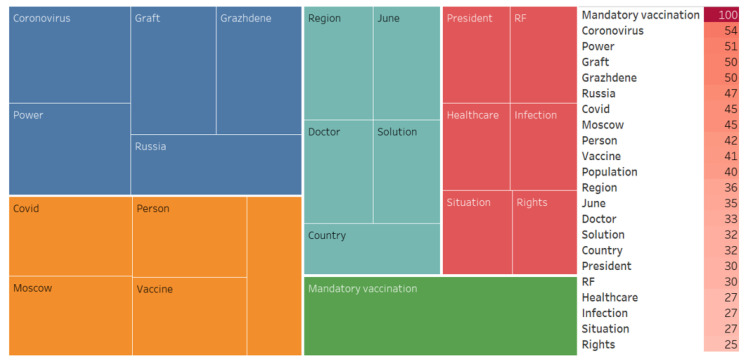
Associative network of the “Mandatory vaccination” stimulus in the negative cluster.

**Figure 8 ijerph-20-00971-f008:**
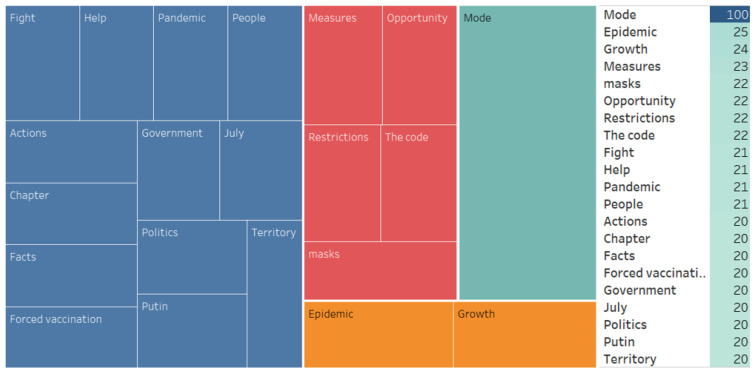
Associative network of the “Distance learning for universities” stimulus in the negative cluster.

**Figure 9 ijerph-20-00971-f009:**
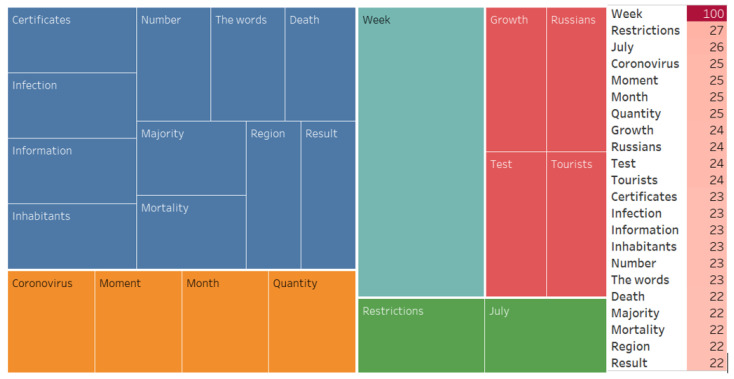
Associative network of the “Non-working days” stimulus in the negative cluster.

**Figure 10 ijerph-20-00971-f010:**
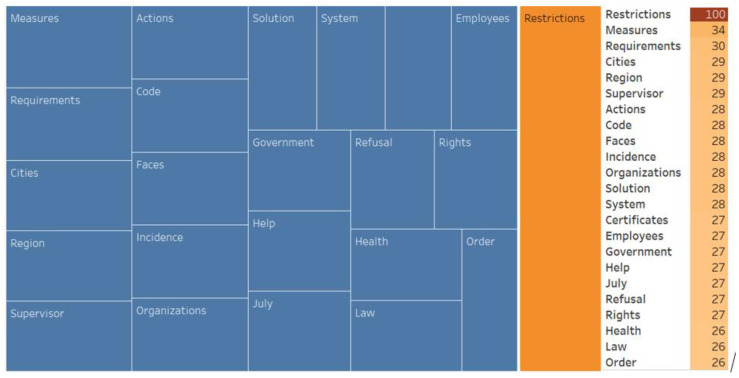
Associative network of the “Quarantine restrictions” stimulus in the negative cluster.

**Figure 11 ijerph-20-00971-f011:**
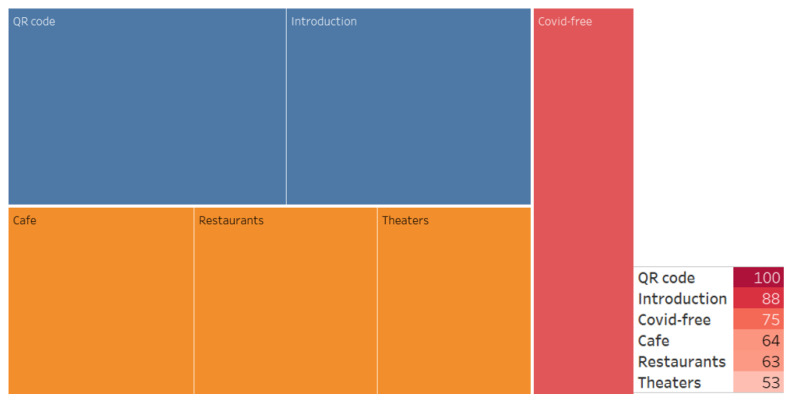
Associative network of the “QR codes” stimulus in the negative cluster.

**Figure 12 ijerph-20-00971-f012:**
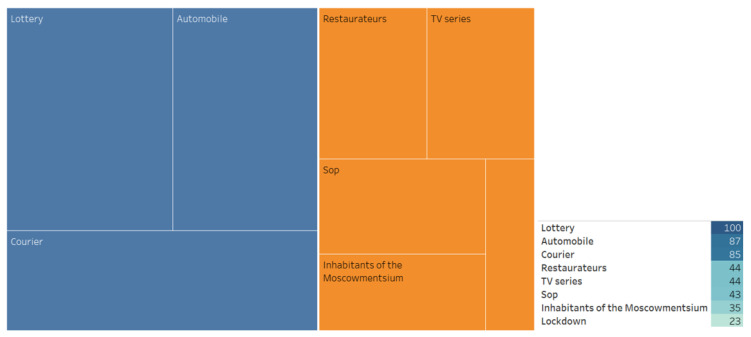
Associative network of the “Vaccination prize drawing” stimulus in the negative cluster.

**Figure 13 ijerph-20-00971-f013:**
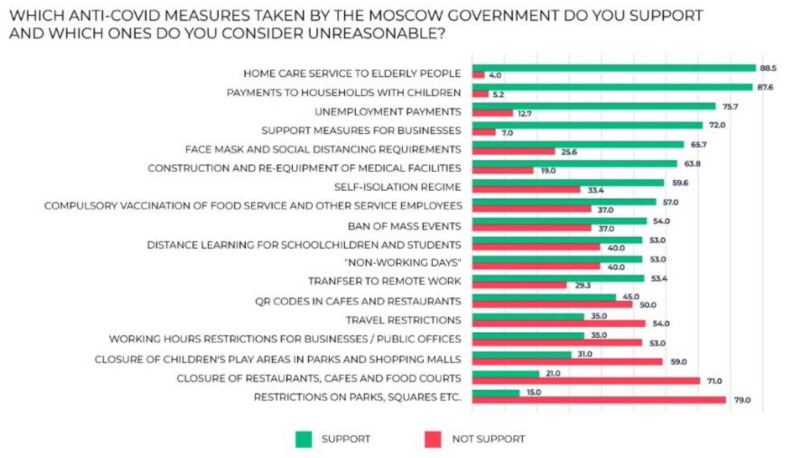
The respondents’ answers to the question: Which anti-COVID-19 measures taken by the Moscow government do you support and which ones do you consider unreasonable?

**Figure 14 ijerph-20-00971-f014:**
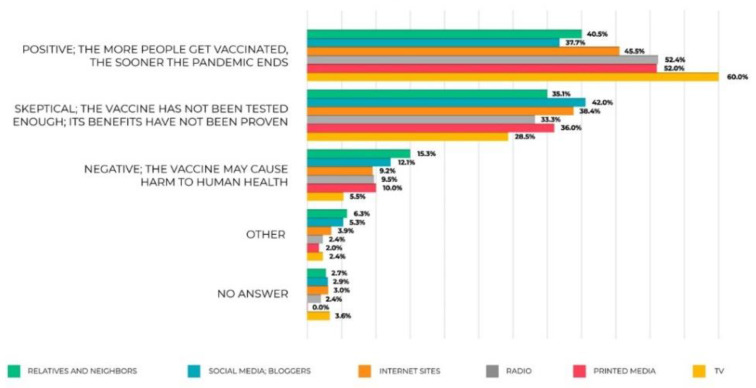
The respondents’ answers to the question: What is your attitude towards vaccination against COVID-19?

**Figure 15 ijerph-20-00971-f015:**
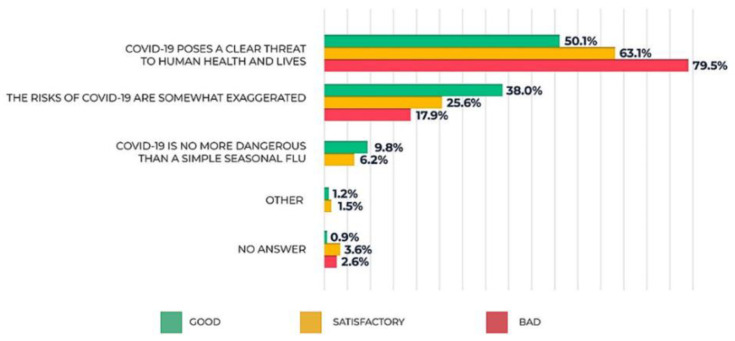
The respondents’ answers to the question: What is your attitude towards the risks the COVID-19 pandemic presents?

**Figure 16 ijerph-20-00971-f016:**
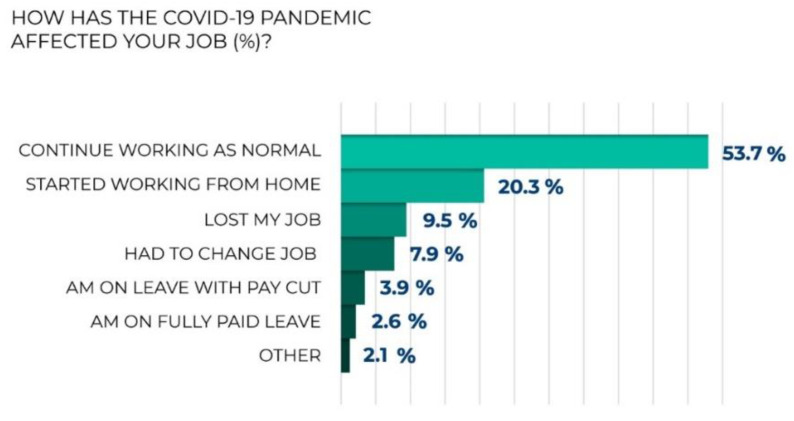
The results of the poll: How has the COVID-19 pandemic affected your job (excluding unemployed respondents)?

**Table 1 ijerph-20-00971-t001:** Data Sets.

Dataset	Authorities’ Actions
Non-Working Days	Declaration of 15 June 2021 to 19 June 2021 non-working days
Quarantine Restrictions	Closure of food courts and children’s playrooms in shopping malls, nighttime curfew for cafes and entertainment enterprises, restrictions in parks
Vaccination Prize Drawing	Drawing of cars for recently vaccinated citizens
Distance Learning for Universities	Transition to online learning for universities
Mandatory Vaccination	Mandatory vaccination for various groups of employees, temporary suspensions for non-vaccinated in several economic sectors
QR codes	COVID-19 free zones in cafes and restaurants, introduction of QR codes for visiting cafes, restaurants, theatres etc.

## Data Availability

Publicly available datasets were analyzed in this study.

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
