# Peer review of "COVID-19 and Public Health: Analysis of Opinions in Social Media"

_ijerph, 2023, doi:10.3390/ijerph20020971_

Round 1
Reviewer 1 Report
The author covered a very current topic, but at the same time I would like to support his work with a few points:
- the literature research: the processed sources are relevant and correct, but at the same time it would be worthwhile to explore the international literature more thoroughly, since representatives of many scientific fields have written about the effects of social media (the author himself emphasizes that he investigated the issue from a multidisciplinary point of view, however, the literature representation of this is missing in his writing ),
- the methodology is not clear, it would be advisable to formulate the research methodology (presented later) in detail already at the beginning of the work,
- similarly, I believe that it would be advisable to specify the hypotheses at the beginning of the study and formulate (final) conclusions accordingly.
The Author investigates the given question in the case of one country/city (in some cases Moscow). It would also be advisable to indicate whether similar studies have been carried out elsewhere and, if so, with what results (the international character should be somewhat strengthened).
Reviewer 2 Report
Dear Authors,
On the whole, I have found the manuscript very interesting and gap-filling, but a few comments were made on the comparison of the results of different methods (text analysis and sociological survey), performed in different timeframes of the investigated third wave of the COVID-19 pandemic. I would suggest expressing your standpoints regarding the below-listed comments and reflections. It would be great to see that infodemic has only an overestimated effect on public opinion measures by your sociological survey.
Charts and figures should be developed. Some of them are too fragmented and/or colouring/colour codes should be unified or corrected.
Detailed comments:
Row 32-33:
It is evident, that elderly people and those suffering from chronic disease(s) or who have impaired immune systems are more vulnerable and susceptible to COVID-19 or any other infection(s). But this may mean comparing medicines and equipments to health and immunity. (vaccines are also preventive substances preparing/stimulating immunity)
Amendment or reference may be relevant.
Row 34-36:
Which reference or result may confirm this statement?
Would this be the long-term effect or the immediate reaction to public health measures? Statements, (rows 49-56) according to the 7th reference, emphasize that responses to actions resulting in great changes and inconveniences are often negative.
Row 43:
letter i is doubled (Epidemiics)
Figure 2:
Colouring/colour codes should be unified on the chart. I would recommend using the same colour code to mark media types on the chart and in the table. Or the use of another type of pie chart should be considered.
Figure 3:
The stories category is missing from this listing.
Row 251-253:
The timeshift can be relevant to exclude this kind of emotional reaction, but also can mitigate the burden of critical responses, since, in the ascendant phase of the wave, the first reaction to public health measures may result in higher resistance and negative attitude in citizens, than at the end of the wave, when the given measures have already achieved their primary goals. In other words, the opinions expressed in social media may be changed positively and not heated so emotionally in the descending part of the third wave. Furthermore, the acceptance of authorities' activities can be higher in the third wave compared to the first one.
The figure of Digital content analysis: A letter g is missing: Content clustering.
Figure 4:
The chart is too fragmented and the colouring is confusing. Using three different colours for the three different sentiments would be better. Grouped bar graph can also be informative dividing results into three main groups by sentiments.
Figure 5:
There are too many items of the same or very similar size, which makes the chart not enough informative with this colouring. The letters are too small.
This comment is relevant in case of the Figures Nr 5 - 10 as well.
Figure 13: Colour codes of the chart are wrong
Row 587: redundant hyphen (relat-ed)
Row 588: redundant hyphen (pro-motion)
Row 591: redundant hyphen (re-actions)
Row 595: redundant hyphen (pow-erful)
Row 596: redundant hyphen (crit-icism)
Row 598: redundant hyphen (ex-cessive)
Row 601: redundant hyphen (physi-cal)
Row 604: redundant hyphen (af-ter)
Row 606: redundant hyphen (possi-bility)
Row 611: redundant hyphen (vaccina-tion)
Row 621-623:
Applying different methods to analyse public opinion is relevant. The measured difference may be influenced by the different phases of the third wave of the COVID-19 pandemic. There is no overlapping between the investigated timeframes, which may contribute to divergence in public opinion measured by text analysis and sociological survey. As already in the methods have been mentioned, the survey was performed in the descending part of the third wave, when public opinion is not influenced so emotionally, then in the ascendant part of the wave, when authorities initiated actions influencing everyday life seriously. But the opinion of social media may also change or, what is more likely, the frequency of negative messages may decrease in the descending part.
Row 624: redundant hyphen (vac-cination)
Row 630: redundant hyphen (criti-cisms)
Row 634: redundant hyphen (de-velopment)
Reviewer 3 Report
The authors present interesting work on analysing publicly expreseeed reactions towards governmental policies during the third wave of the COVID-19 pandemic in Russia.
Some sections however (mainly the introduction), are not very clearly expressed and could be significantly imporved by revising the linguitic syntactical structure of the document. The discussion and conclusions on the other hand are very well written, with a minor problem in the conlcusions being the hyphenation and some strange word spacings that need to be corrected.
The visualisations in the results need to be better explained and clarified so that the reader can easily understand the meaning they convey.
The duration that is examined (45 days) is a bit limited to be able to consider the results very representative, but it does provide the genral feeeling of the public opinion.
The methodology is explained, but not to a level of detail that it could be easily reproduced by other researchers analysing other data and problems.
Reviewer 4 Report
It is an interesting topic which tried to get the perceptions of the peaople around the government actions during Covid-19 pandemic. However, some comments are provided to enrich the manuscript.
1. the author stated that based on the purpose of the study six text corora were formed. The author is recommended to discuss the criteria to select these six corpora and the reasons he/she believe they are the most important ones that deserve evaluation and consideration.
2. the text has some minor typos and grammatical errors which should be collected for example, "data was" should be changes to "dara were" as the word "data" is plural. The author is recommended to proof read the manuscript.
3. The author used different analysis techniques for each part of the collected data. But justification was not provided why each one of them is selected out of all other analysis techniques. Thus, the author is recommended to discuss the validity and reliability of each one of the analysis techniques used and elaborate why they are the selected one out of similar analysis techniques and instruments.
4. In the last paragraph before the 2.3 section, the author claimed using random, sampling while he/she immediately stated using age and gender criteria in selection. It is confusing that the author used random sampling or purposive sampling. It should be clarified.
5. Digital content analysis was described through figures without any explanation; whereas complementary explanation is required to make it reader friendly.
6. There is no in-depth discussion section to compare and contrast the findings with relevant/similar studies done by other researchers. The author should use critical thinking in discussing the similarities or differences between his/her own studies and the results found by others; moreover, the author should provide the potential reasons for his/her findings.
7. refereces need minor adjustment to be compatible with the latest reference style.
